# Energetic equivalence underpins the size structure of tree and phytoplankton communities

Daniel M. Perkins[1], Andrea Perna[1], Rita Adrian[2], Pedro Cermeño [3], Ursula Gaedke[4], Maria Huete-Ortega[5], Ethan P. White[6,7,8] & Gabriel Yvon-Durocher [9]

The size structure of autotroph communities – the relative abundance of small vs. large individuals – shapes the functioning of ecosystems. Whether common mechanisms underpin the size structure of unicellular and multicellular autotrophs is, however, unknown. Using a global data compilation, we show that individual body masses in tree and phytoplankton communities follow power-law distributions and that the average exponents of these individual size distributions (ISD) differ. Phytoplankton communities are characterized by an average ISD exponent consistent with three-quarter-power scaling of metabolism with body mass and equivalence in energy use among mass classes. Tree communities deviate from this pattern in a manner consistent with equivalence in energy use among diameter size classes. Our findings suggest that whilst universal metabolic constraints ultimately underlie the emergent size structure of autotroph communities, divergent aspects of body size (volumetric vs. linear dimensions) shape the ecological outcome of metabolic scaling in forest vs. pelagic ecosystems.

[1] Department of Life Sciences, Whitelands College, University of Roehampton, London SW15 4JD, UK. [2] Leibniz Institute of Freshwater Ecology and Inland Fisheries (IGB), Department of Ecosystem Research, Müggelseedamm 301, 12587 Berlin, Germany. [3] Institute of Marine Sciences (ICM-CSIC), Passeig Marítim de la Barceloneta, 37–49, 08003 Barcelona, Spain. [4] Institute for Biology, University of Potsdam, 14469 Potsdam, Germany. [5] Oroboros Instruments, Schöpfstraße 18, 6020 Innsbruck, Austria. [6] Department of Wildlife Ecology and Conservation, University of Florida, Gainesville, FL 32611, USA. [7] Informatics Institute, University of Florida, Gainesville 32611 FL, USA. [8] Biodiversity Institute, University of Florida, Gainesville 32611 FL, USA. [9] Environment and Sustainability Institute, University of Exeter, Penryn, Cornwall TR10 9EZ, UK. Correspondence and requests for materials should be addressed to D.M.P. (email: daniel.perkins@roehampton.ac.uk) or to G.Y-D. (email: G.Yvon-Durocher@exeter.ac.uk)

A striking difference between aquatic and terrestrial realms is the size of their dominant autotrophs[1]: approximately 11 orders of magnitude in body mass separate unicellular algae and multicellular vascular plants[2]. Understanding whether the structure of autotroph communities is shaped by common underlying mechanisms is fundamental to efforts towards modeling primary production[3,4], understanding constraints on the availability of energy to higher trophic levels[5,6], and determining whether ecological systems are governed by general laws[7]. The individual size distribution (ISD)—the frequency distribution of individual body sizes in a community—describes how energy and resources in an ecosystem is partitioned among individuals[8] and is one of the most extensively studied patterns in aquatic and terrestrial ecology[2,9–13]. Ecological communities comprise many small and few large individuals and ISDs have often been characterized using a power-law[8], where the frequency of individuals of body size, $M$, follows a function of the form, $f_M \propto M^\lambda$, where $\lambda$ the exponent is negative (i.e., $\lambda < 0$).

Metabolic scaling theory (MST) proposes that the decline in the number of large individuals can be explained by the sub-linear scaling of metabolic rate with body mass[14,15], and by trade-offs between the number of individuals and the amount of resources that each individual can acquire in an ecosystem with finite resources[15–17]. Consequently, ecosystems can support (relatively) few large individuals, which require more resources to sustain their metabolism. This concept, termed Energetic Equivalence[8], yields the expectation that the power-law exponent of the ISD should be inversely proportional to the metabolic scaling exponent[17].

Since metabolic rates tend to scale as $M^{3/4}$ for large vascular plants[15,17–19] and eukaryotic algae[15,19–22], the theoretical expectation is that the ISD follows a power-law with an exponent approximating -¾ in both tree and phytoplankton communities. This notion has received some empirical support[12,17], though many counter examples also exist[10,11,23,24]. Previous tests of energetic equivalence have used a wide variety of aggregation methods[8], statistical techniques[25,26], and measures of body size[16,27] for assessing the scaling of abundance and body size in tree and phytoplankton communities, severely limiting efforts to reconcile these scaling laws across aquatic and terrestrial realms.

We carry out the first standardized analysis of individual size distributions from a global dataset of 2062 tree and phytoplankton surveys from 242 terrestrial and 95 aquatic (freshwater and marine) sampling locations (Fig. 1 and Supplementary Table 1). We first establish the general form of ISDs for both tree and phytoplankton communities by testing different power-law distributions (power-law, bounded power-law and power-exponential; Supplementary Table 2), as well as alternative distributions, using a maximum likelihood approach (Methods). We then determine the coefficients of the best-fitting ISD at each location and test for macroecological differences in size structure between the two taxonomic groups (Methods). Our results reveal both fundamental differences as well as striking similarities in the mechanisms that underpin the emergent size structure of aquatic and terrestrial autotroph communities.

## Results

### Characterizing the form of individual size distributions.
Power-law distributions provided a good fit to the tree and phytoplankton data when fitted on sizes above a site-specific threshold (Fig. 1 and Supplementary Table 3). The empirical ISD was statistically indistinguishable from the theoretical distribution of the best-fitting power-law model in 87% of tree communities and 83% of phytoplankton communities (Supplementary Data 1). The bounded power-law was the best-supported distribution in

tree communities while the bounded power-law and power-exponential distribution were approximately equally well supported in phytoplankton communities (Table 1). The preference of the bounded power-law (and power-exponential) distribution over the (unbounded) power-law occurs because of curvature in the tails of the size distributions (Fig. 1) and implies some inherent maximum size that individuals can attain[26].

**Comparing power-law exponents**. Comparing ISD exponents ($\bar{\lambda} + 1$; Methods) revealed significant differences between tree and phytoplankton communities (two-sample $t$-test: $t = 11.39$, df = 329, $P < 0.001$) with a mean for trees of −0.47 (95% confidence interval: −0.49 to −0.44; Fig. 2a) and phytoplankton of −0.79 (95% confidence interval: −0.85 to −0.73; Fig. 2b). Estimates of the mean exponent were robust to the exclusion of sites where empirical ISDs differed from the theoretical distribution: trees −0.47 (95% confidence interval: −0.50 to −0.44) and phytoplankton −0.76 (95% confidence interval: −0.81 to −0.70). Altogether, these results highlight striking differences in the size structure of the dominant autotrophs in aquatic and terrestrial ecosystems, with proportionately fewer individuals of large mass found in phytoplankton communities.

**Phytoplankton time-series analyses**. A key assumption in deriving scaling laws linking individual metabolism to community size structure is that communities are at demographic and resource steady state[15,16] so that, on average, the total rate of resource-use equals the rate of resource supply, birth rates approximate death rates, and a stable distribution of ages and sizes exists[16]. The turnover of phytoplankton community composition is rapid in response to environmental variability owing to their small size and high capacity for dispersal. Consequently, point measurements of the ISD at a given location only provide a snapshot of community structure, which may deviate from steady state depending on the local disturbance history. In order to test for the effects of temporal variation, we leveraged extensive time-series data of four marine and two limnetic freshwater stations. We aggregated data over multiple years for each station (Supplementary Table 1) to build up a picture of the average composition of phytoplankton communities in the long-term (Methods). We compared these temporally aggregated ISD exponents to the average exponents based upon all point measurements of the ISD within each station (Supplementary Fig. 2).

The ISD exponents after temporal aggregation were statistically indistinguishable from the ISD exponents derived from point measurements (two-sample $t$-test: $t = -1.26$, df = 10, $P = 0.237$; Supplementary Fig. 2), or from exponents observed for the spatial phytoplankton surveys (two-sample $t$-test: $t = -1.18$, df = 93, $P = 0.239$; Fig. 2b). Consequently, the estimated ISD exponents for temporally aggregated phytoplankton communities ($\bar{\lambda} + 1 = -0.65$; 95% confidence interval: −0.83 to −0.48; Fig. 2c) were also significantly larger than for tree communities ($t = 2.15$, df = 246, $P = 0.032$), indicating that the contrasting size scaling between these groups cannot be explained by differences in demographic equilibria between unicellular algae and vascular plants.

## Discussion
Here, we provide the first unified statistical analysis of the size structure of the planet's dominant autotrophs[3]. Our results reveal fundamental differences in the individual size distribution (ISD) between trees and phytoplankton with proportionally fewer individuals of large mass found in phytoplankton communities. However, these divergent patterns in autotroph size structure

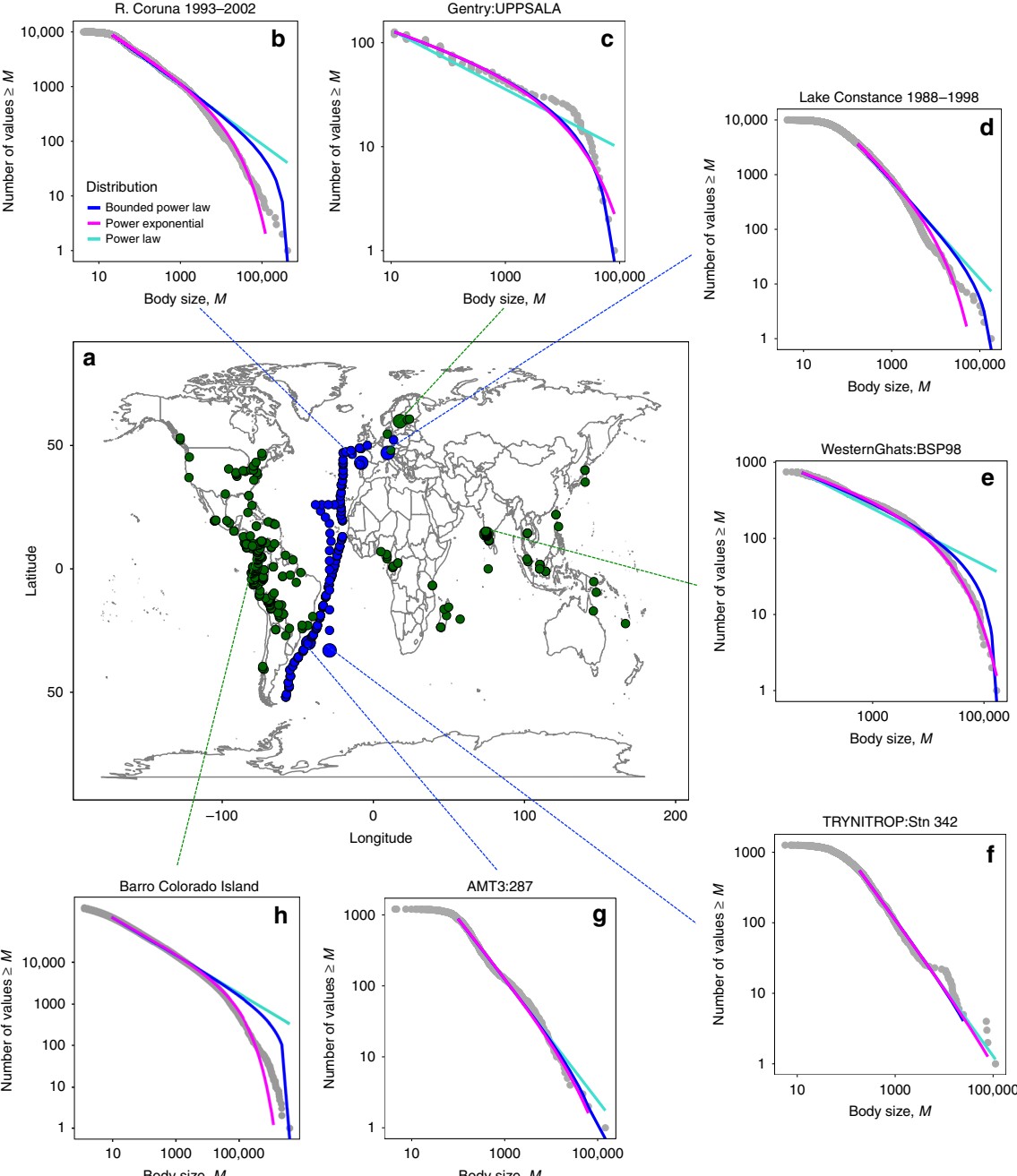

**Fig. 1** Global distribution of survey sites including representative individual size distributions. **a** Green and blue data points denote terrestrial and aquatic sampling locations, respectively. **b**–**h** A subset of rank-frequency plots which gives, on $\log_{10}$ axes, the rank of body size, $M$, ($\mu m^3$ for phytoplankton and $D^{8/3}$ for trees, where $D$ is tree stem diameter in cm) and the number of values $\geq M$. The bounded power-law (blue fitted line) was generally the best-supported distribution for both tree and phytoplankton communities (Table 1), out-performing the (unbounded) power-law or power-exponential distributions (turquoise and magenta fitted lines, respectively)

**Table 1 Identifying the best-fitting individual size distribution**

| Dataset | Number of sites | Power-law | Bounded power-law | power-exponential | Mean ISD exponent + 1 | 95% confidence intervals |
|---|---|---|---|---|---|---|
| Trees | 242 | 0.14 | 0.86 | 0.42 | −0.47 | −0.49 to −0.44 |
| Phytoplankton (spatial) | 89 | 0.33 | 0.70 | 0.64 | −0.79 | −0.85 to −0.73 |
| Phytoplankton (temporal) | 6 | 0.17 | 0.17 | 0.83 | −0.66 | −0.82 to −0.50 |

The proportion of occasions that each form of power-law distribution (power-law, bounded power-law, and power-exponential) was ranked among the best models (see Methods) is given for each dataset. The mean ISD exponent (and 95% confidence intervals) was derived for each dataset from the best-fitting power-law distribution at each location.

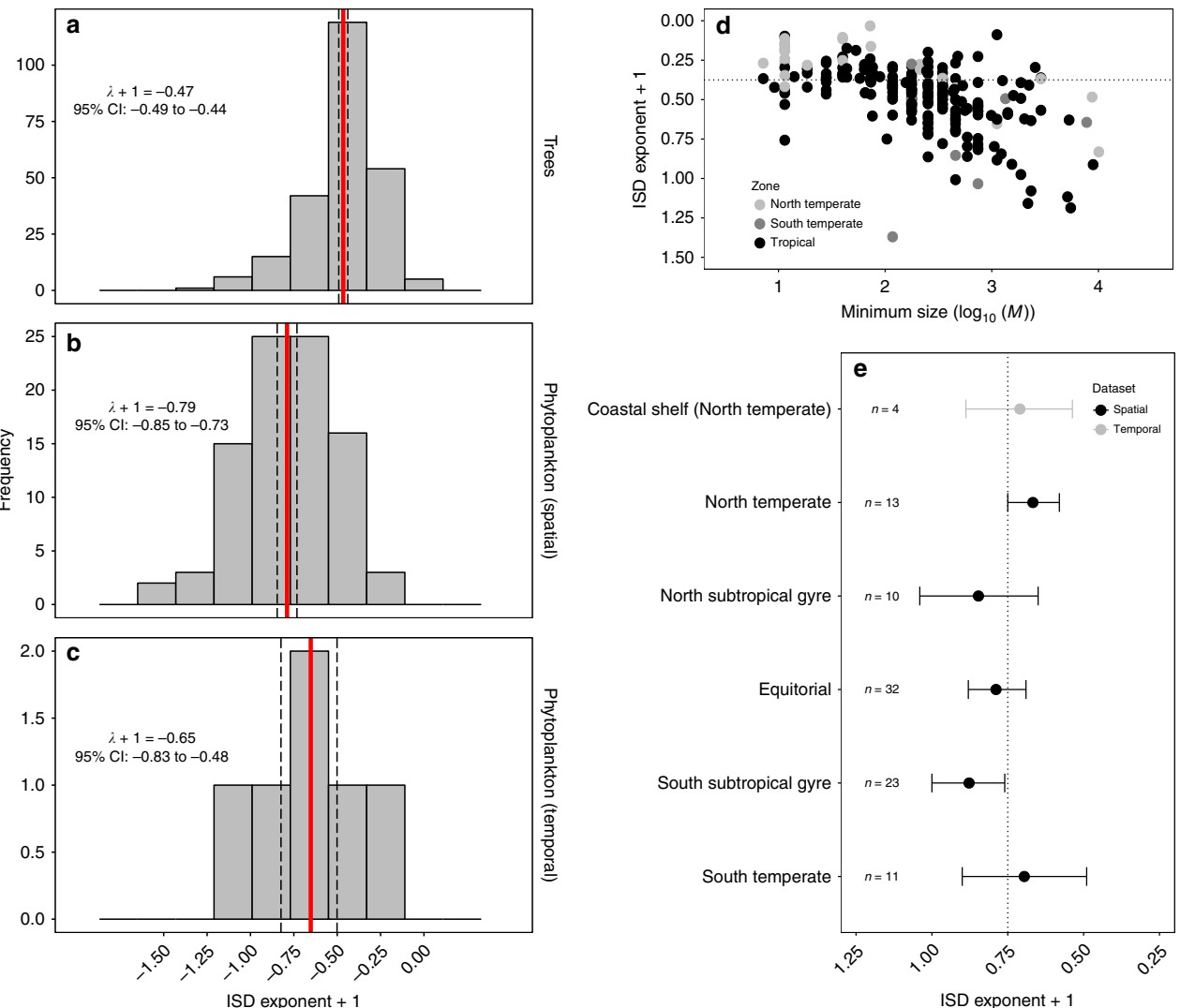

**Fig. 2** Individual size distributions differ between phytoplankton and trees and under different environmental contexts. **a–c** Phytoplankton individual size distribution (ISD) exponents are significantly larger compared to tree communities highlighting that proportionally more small, relative to large, individuals are found in phytoplankton communities. The mean ISD exponent ($\bar{\lambda} + 1$) and 95% confidence intervals are indicated in each panel by solid and dashed vertical lines, respectively. **d** Tree ISD exponents become more negative, and deviate increasingly from the metabolic scaling prediction (dashed line), as the minimum body size in the fitted power-law distributions increases. Large minimum size values signify significant deviation from the general power-law scaling function among small size classes and likely reflect recruitment limitation due to external disturbances. **e** A general ISD exponent ($\bar{\lambda} + 1 \approx -0.75$) is supported among nearly all the major oceanic regions across the globe, consistent with metabolic scaling theory (dashed line). Error bars represent 95% confidence intervals

appear to be underpinned by common metabolic mechanisms, which we discuss in the following paragraphs.

Our results reveal a general ISD exponent ($\bar{\lambda} + 1 \approx -0.75$) for very different pelagic ecosystems across the globe (Fig. 2e) that is independent of the scale of spatial or temporal aggregation. This average ISD exponent is consistent with expectations from metabolic scaling theory (MST) and energetic equivalence[8], assuming a three-quarter power scaling of metabolic rate with body mass[15] and equivalence in resource use among logarithmic mass classes in a community[15,28]. The theoretical three-quarter-power scaling of metabolism with body mass for phytoplankton is in agreement with a number of empirical studies[15,19–22] but contrasts with some studies that have reported isometric scaling exponents for phytoplankton metabolic rates both in the field[24,29] and in the laboratory[30]. These discrepancies likely arise, in part, because the latter studies included smaller size fractions (pico-plankton, which include prokaryotic autotrophs), than that

studied here: when only species larger than 100 μm³ are considered (approximately the lower bound of the fitted distributions in this study; Fig. 1), the scaling of metabolism approximates ¾[30].

The average ISD exponent for tree communities ($\bar{\lambda} + 1 = -0.47$, 95% confidence intervals: $-0.49$ to $-0.44$), differs significantly from expectations based on energetic equivalence among logarithmic mass classes, indicating that tree communities have a greater proportion of individuals with large body mass. Metabolism tends to scale with body mass with an exponent approximating ¾ for vascular plants (ranging from 0.1 to 100 cm in diameter[18,31]). Although a larger exponent of ~ 1 is found among seedlings and saplings[32], we are unaware of any data that supports a body mass-metabolism scaling exponent for trees of ~ 0.5. However, the average scaling exponent of the ISD derived using body mass ($\bar{\lambda} + 1 = -0.47$) is similar to the scaling coefficient ($\bar{\lambda} + 1 = -0.375$) that would be expected from a mass-based transformation of the ISD using linear tree diameter

classes[16,27,33]. Indeed, further inspection of the data reveals that deviations from the expected ISD exponent occur primarily among tree communities with a 'large' minimum size above which power-law scaling is supported ($\log_{10}$ minimum body size > 2.5 $cm^{8/3}$; see Fig. 2d and Methods). This is consistent with the idea that forests with few small individuals, and those where the distribution of sizes among small size classes deviates from the general power-law scaling, reflect recruitment limitation[13] due to external disturbances[9,11,23] such as fire, climate extremes and size-dependent herbivory. These communities therefore do not follow the expected ISD scaling because they violate the inherent assumptions of demographic steady state in metabolic scaling theory[16]. Tree communities with a small minimum body size above which power-law scaling is supported have scaling exponents that are statistically indistinguishable from the expectations ($\bar{\lambda} + 1 \approx -0.375$) based on energetic equivalence among logarithmic diameter classes ($\bar{\lambda} + 1 = -0.39$, 95% confidence intervals: $-0.42$ to $-0.36$). Thus, whilst tree communities have ISDs that deviate from energetic equivalence based on body mass, ISD exponents based on diameter ($\bar{\lambda} + 1 \approx -2$) are indeed consistent with three-quarter-power metabolic scaling and equivalence in resource use among diameter classes[16,27].

A key difference between tree and phytoplankton communities concerns mixotrophy, the ability of some photosynthetic organisms to also ingest and assimilate living prey as a means of supplementing nutrient acquisition. While largely absent in forests, mixotrophy is ubiquitous in pelagic ecosystems[34] and is suggested to exert a strong influence on the size structure of plankton communities at the global scale[35]. We are unable to quantify any putative impacts of mixotrophy directly[36] in this study given the data we have at our disposal. However, for the vast majority of chloroplast-bearing protists that contribute significantly to primary production in pelagic ecosystems, energy metabolism is primarily driven by light and photosynthesis[37,38]. Consequently, all of the taxa included in our analyses are primary producers—a small but variable fraction may access additional nutrition to fuel photosynthesis via phagotrophy when inorganic nutrients are scarce—but ultimately all use sunlight to fix carbon at the base of the food chain. Recent theoretical work[35] suggests that the mixotrophic contribution to primary production is greater in the nutrient depleted oligotrophic regions of the ocean and therefore we expected to observe less negative ISD exponents (i.e., shallower size spectra scaling with more large individuals) under oligotrophic conditions if they have access to alternative energy sources via mixotrophy. In contrast, when analyzing latitudinal variation in the ISD exponent (Fig. 2e), we find that more negative exponents occur in the oligotrophic oceanic regions (subtropical gyres) owing to a predominance of small size classes and few large individuals[10]. We hypothesize that the convergence in the size scaling exponent of phytoplankton communities with expectations based on energetic equivalence suggest that these primary producer communities are primary fueled by sunlight and that whilst mixtrophy may allow some members to access additional nutrient pools under certain conditions, it does not systematically violate the assumptions of energetic equivalence based on a common energy source (e.g., sunlight).

Finally, it is important to consider that other taxonomic groups, not studied here, contribute to the total autotrophic production in forest and pelagic ecosystems. In forests, shrubs and herbaceous plants are abundant at the lower end of the size spectrum[39] whilst picophytoplankton (including cyanobacteria and picoeukaryotes) contribute significantly to primary production in pelagic ecosystems, especially in oligotrophic waters[40]. We are unaware of comparable large-scale datasets comprising the

size and abundance of shrubs and herbaceous plants so it is uncertain if the general scaling pattern we observe for trees (> 1 cm in diameter) extends down the size spectrum to these groups. Some evidence suggests that the individual size distribution for the entire autotrophic component of forests could be a discontinuous function with few species filling the size gap between shrubs and trees[39]. In oligotrophic pelagic systems where cyanobacteria are abundant, a transition towards a steeper decline in abundance with body size might be expected[24] given the super-linear body mass-metabolism scaling for prokaryote species[41,42] and equivalence in energy use among logarithmic mass classes.

By carrying out a unified statistical analysis using global datasets, our study reveals both fundamental differences as well as striking similarities in the mechanisms that underpin the emergent size structure of the planet's dominant autotrophs. Our results intimate that the stark differences in the physical environment experienced by vascular plants in the terrestrial realm and unicellular phytoplankton in the water column may mean that different aspects of organism size (i.e., linear dimensions of a tree vs. volumetric dimensions of a unicellular alga) play important roles in governing population and community dynamics in forest and pelagic ecosystems. Nevertheless, our findings make a strong case for the existence of unified constraints that govern the size structure of aquatic and terrestrial autotrophs based on commonalities in the size scaling of metabolism[15,43] and trade-offs between the number of individuals, and the amount of resources that each individual can sequester in ecosystems with finite resources. These results imply a relatively simple scaling-up of resource use across levels of biological organization that could facilitate improvements in how models of global biogeochemical cycles represent autotroph biodiversity.

## Methods

**Data compilation.** Tree community data included individual size measurements collated from the Gentry transect dataset[44] and 55 forest plots[45] available with permission (Supplementary Table 1). We used 187 (of the original 226) Gentry sites, where individual stems were surveyed for a standardized area (2 × 50 m transects = 0.1 ha per site;[46]). Each of the 55 forest plots was at least 1 ha in size and the plots analyzed here span four continents of tropical and temperate closed-canopy forests (Supplementary Table 1). All forest transects and plots were fully surveyed, with diameter at breast height (DBH) measured for all individuals above location-specific minimum thresholds (Supplementary Table 1). We calculated body mass from DBH using a general allometric model of plant vascular systems where tree mass ($M$) is proportional to the 8/3-power of stem diameter ($D$), of any size class: $M \propto D^{8/3}$[47]. Empirical relationships are statistically indistinguishable from this theoretical value[2,48–50] and vary little between temperate and tropical forests, although some evidence suggests this relationship predominately holds for small size classes in tropical forests[51]. For the forest plots, where individuals with multiple stems were identified, we adopted the pipe model to combine the records, where $D = (\sum d_i^2)^{1/2}$ and $d_i$'s were the diameters of individual stems[45,52]. The Gentry data does not identify the stems with the individual that they came from making it impossible to back calculate the basal stem diameter for an individual. Consequently, these data have typically been treated as if every stem is a different individual[17,25] and we do so here.

Phytoplankton community data (nano- and microphytoplankton) included 92 open ocean stations, 75 from the Atlantic Meridional Transects (AMT 1–3[53]) and 17 from an additional Atlantic Ocean survey[24]. Phytoplankton time-series data included four temperate coastal stations (Ría de A Coruña[54], Ría de Vigo, Atlantic Iberian Shelf, and L4 English Channel[53]) and two freshwater lakes (Lake Constance[55] and Müggelsee[56]) (Supplementary Table 1). AMT cruises crossed the same regions of the Atlantic Ocean, from 48°N to 50°S, by a similar route but we treated each sampling station as a unique community. Where multiple samples were taken from various depths at each location—often samples were collected from the surface to the bottom of the euphotic layer (Supplementary Table 1)—they were pooled for each community. Time-series data consisted of weekly to monthly surveys of phytoplankton and to assess community size structure over the long-term all surveys were pooled for each location (1–18 years; Supplementary Table 1). Microscopic analyses for all phytoplankton datasets followed standardized procedures: two replicate water samples were preserved in buffered formalin (to preserve calcium carbonate structures) or Lugol's iodine solution and analyses of samples were carried out following 24–48 h sedimentation (Utermöhl

technique), with cells identified to species (or morphotype) level and a subset of taxon measured to calculate cell volume[53–56]. The use of Lugol's solution can results in changes in the size (shrinkage and swelling) of phytoplankton cells[57–59]. However, such effects are highly variable between groups[58] and given the preservation of samples followed a standardized protocol for all datasets, we do not correct for the effects of preservation: we assume any effect will be negligible[58] given the 4–6 orders of magnitude variation in phytoplankton cell volume in this study (Fig. 1).

With the exception of 17 Atlantic Ocean stations where all individual sizes were recorded[24], phytoplankton datasets consisted of taxon-average cell volume and abundance so it was necessary to estimate individual-level size distributions[60]. Assigning a mean taxon-specific size to every individual of that taxon eliminates realistic intra-specific variation so we simulated individuals from compiled available data on the mean body size and variance of 127 freshwater[61] and 243 marine[62] phytoplankton species. We used equivalent spherical diameter (ESD) rather than cell volumes as the measurement of body size, as the former tends to be more normally-distributed within species[62]. The freshwater database was filtered to exclude entries where the number of measurements per species was less than ten per location and where species body size measurements were made in less than five locations[62]. We performed linear mixed effects modeling using the lmer function in the lme4 package in the R statistical platform (v. 3.4.1[63]) to determine the general scaling relationship between the log-transformed mean and standard deviation (SD) of phytoplankton ESD (Supplementary Fig. 1), fitting ecosystem realm (freshwater or marine) as a random effect on the intercept. The continuous ISD was estimated from this strong relationship ($SD[ESD] = 0.144\ ESD^{1.21}$, conditional $R^2 = 0.81$; Supplementary Fig. 1) by randomly sampling individual sizes for each species from a normal distribution with the reported taxon-specific ESD in the original data sources and the standard deviation estimated from the derived equation. To facilitate comparisons with previous studies, ESD values were converted to cell volumes ($\mu m^3$) prior to fitting the individual size distributions. For communities where the number of simulated individuals exceeded 10,000, individuals were sampled (without replacement) to this maximum to reduce computational time when fitting individual size distributions and for comparability with tree datasets.

**Data analysis**. To test metabolic scaling theory (MST) predictions that the individual size distribution (ISD) is a power-law with an exponent approximating −3/4, we adopted the method of Clauset et al.[64] to find both the best-fit minimum size, $x_{min}$, to which a power-law applies and the scaling exponent, $\lambda$, using the plfit function implemented in R (http://tuvalu.santafe.edu/~aaronc/powerlaws/plfit.r). This method minimizes the Kolmogorov–Smirnov (KS) statistic comparing a community's size distribution with a power-law distribution by iteratively selecting increasing values of $x_{min}$. However, larger $x_{min}$ values will reduce the community's sample size, as only individuals with a size above $x_{min}$ are included in the power-law fitting and this can return erroneously large estimates of power-law exponents. To prevent this, we truncated the search over $x_{min}$ values before the finite-size bias becomes significant and skip $x_{min}$ values with finite-size bias > 0.05 (i.e., 5% error around the theoretical distribution) and we selected the first value of $x_{min}$ within the 25th percentile of the optimised $x_{min}$.

We fitted a range of power-law distributions (power-law, bounded power-law, and power-exponential), as well as alternative distributions (log-normal and Weibull) to size data above the selected $x_{min}$ for each community, using published R functions[64,65]. Rank-frequency plots were used to visualize the resulting fit (Fig. 1), which gives, on logarithmic axes, the rank of body size, $M$, (the number of values ≥ $M$) against the value of $M$[65]. We used a standard maximum likelihood approach[66] and compared models based upon log-likelihoods and Akaike's Information Criterion (AIC) with lower AIC scores representing models with better fit to the data[67]. To identify the best-fitting distribution across communities, models were ranked according to differences from the best-performing model, ΔAIC, and substantial support for a model was determined from the proportion of occasions a model was ranked best or within < 2 ΔAIC[67]. For each community, the probability that the empirical ISD differed from the theoretical distribution for the best-fitting power-law model was calculated from the KS statistic. It is important to note, however, with a large number of observations even small deviations from the theoretical distribution can be detected[64].

To compare the size structure of tree and phytoplankton communities, we extracted the value of the exponent from the best-fitting power-law distribution for each community. The power-law distribution is a transformed version of the abundance spectrum which is commonly used in aquatic ecology[26], but captures the same information in different form: the ISD exponent, $\lambda$, is equivalent to the abundance spectrum slope −1[25]. To aid comparisons with previous work and test MST predictions, we added 1 to ISD exponents and report $\lambda + 1$[28]. We focus on the macroecological trends in exponents as a function of ecosystem realm and calculate the 95% confidence intervals around the global average exponent for phytoplankton and tree communities to assess support for MST predictions. We adopted this approach rather than testing the prediction for each community since sample sizes varied between communities and hence the confidence intervals around fitted exponents would likely be wide when samples sizes were small, and consequently we would be more likely over-report the number of communities that follow the predictions[13]. Two-sample $t$-tests were used for testing for differences in the ISD exponents between datasets (trees, phytoplankton-spatial and

phytoplankton-temporal) and (normal) 95% confidence intervals around estimates of means were determined by bootstrapping ($n = 10,000$) using the boot package in R. We removed three AMT stations as these were outliers in the analysis (Supplementary Data 1).

**Reporting summary**. Further information on experimental design is available in the Nature Research Reporting Summary linked to this article.

## Data availability

The summary data used to generate Table 1 and Fig. 2 are available in Supplementary Data 1. The analysis R code, as well as a subset of the analyzed data, is archived in a Figshare public repository (https://figshare.com/s/013fba909417e89fe7e1). The data included in the deposit are specifically intended for the replication of the analysis procedure. Researchers interested in using the data for purposes other than replicating our analyses are advised to obtain the raw data from the original sources cited here, as other useful information from the original data might not be included. A reporting summary for this article is available as a Supplementary Information file.

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

## Acknowledgements

We would like to thank Xiao Xiao for constructive discussions on earlier drafts. This article was made possible by support from the Natural Environment Research Council (NERC) through the Tansley Working Group 'PerPoce Planet Earth, Planet Ocean' G.Y-D was supported by an European Reasearch Council starting grant (ERC StG 677278 TEMPDEP). We thank the numerous people, organizations, and funding bodies who contributed to the collection of data. R.P. Harris provided the AMT and English Channel data. AMT data collection was supported by the UK Natural Environmental Research Council through the Atlantic Meridional Transect consortium (NER/O/S/2001/00680). Data from the Ría de A Coruña and the Atlantic Ocean stations from the TRYNITROP project were obtained thanks to the funding provided by Spanish Ministry of Education and Science (MEC) through the grants CTM2004-05174-C02 and CTM2008-03699, as well as the by the Instituto Español de Oceanografía through their program RADIALES. Data collection from the Ría de Vigo was supported by projects UE MAST-CT90-0017 and DYBAGA MAR99-1039-C02-01 awarded to F.G. Figueiras. The Müggelsee data are part of the Long-Term Research program of the IGB. R.A. acknowledges support via the the French Foundation for Research on Biodiversity (FRB) through its synthesis center, CESAB (http://www.cesab.org/), and the John Wesley Powell Center for Analysis and Synthesis (https://powellcenter.usgs.gov/). Nathan G. Swenson provided data for wood density in Luquillo forest plot. The Serimbu (provided by T. Kohyama), Lahei (provided by T.B. Nishimura), and Shirakami (provided by T. Nakashizuka) datasets were obtained from the PlotNet Forest Database. The ACA Amazon (provided by N. Pitman) and DeWalt Bolivia (provided by S. DeWalt) datasets were obtained from SALVIAS. The BCI forest dynamics research project was made possible by National Science Foundation grants to Stephen P.Hubbell: DEB-0640386, DEB-0425651, DEB-0346488, DEB-0129874, DEB-00753102, DEB-9909347, DEB-9615226, DEB-9615226, DEB-9405933, DEB-9221033, DEB-9100058, DEB-8906869, DEB-8605042, DEB-8206992, DEB-7922197, support from the Center for Tropical Forest Science, the Smithsonian Tropical Research Institute, the John D. and Catherine T. MacArthur Foundation, the Mellon Foundation, the Small World Institute Fund, and numerous private individuals, and through the hard work of over 100 people from ten countries over the past two decades. The UCSC Forest Ecology Research Plot was made possible by National Science Foundation grants to Gregory S. Gilbert (DEB-0515520 and DEB-084259), by the Pepper-Giberson Chair Fund, the University of California, and the hard work of dozens of UCSC students. These two projects are part of the Center for Tropical Forest Science, a global network of large-scale demographic tree plots. The Luquillo Experimental Forest Long-Term Ecological Research Program was supported by grants BSR-8811902, DEB 9411973, DEB 0080538, DEB 0218039, DEB 0620910 and DEB 0963447 from NSF to the Institute for Tropical Ecosystem Studies, University of Puerto Rico, and to the International Institute of

Tropical Forestry USDA Forest Service, as part of the Luquillo Long-Term Ecological Research Program. The U.S. Forest Service (Dept. of Agriculture) and the University of Puerto Rico gave additional support. E.P. White acknowledges support from the Gordon and Betty Moore Foundation's Data-Driven Discovery Initiative through Grant GBMF4563 and by the National Science Foundation through grant 0953694.

## Author contributions

D.M.P and G.Y.-D. conceived the study. E.P.W, R.A, P.C, U.G., and M.H.-O. contributed data; D.M.P, A.P., and G.Y.-D. analyzed the data; D.M.P. and G.Y.-D. wrote the manuscript and all authors contributed to revisions.

## Additional information

**Competing interests:** The authors declare no competing interests.

