## [Peer Review File · Nature Communications]

Reviewers' comments:

Reviewer #1 (Remarks to the Author):

Perkins et al compare allometric scaling relationships between phytoplankton and tree data. The paper is generally well written, but there seems to be a severe conceptual problem in terms how the two very different biomes are compared with regard to the life span of their organisms.

The scaling relationships addressed here are emergent properties of processes at the level of ecological communities (e.g. resource competition among competing taxa is one such key process). "a community is a group or association of populations of two or more different species occupying the same geographical area and in a particular time" (from wikipedia for convenience). Phytoplankton communities change rapidly due to short generation times (app 1 day) and fast dynamics of their environment (consider e.g. difference between spring bloom and summer stratification situation, see PEG model, Sommer et al). The community is an emergent property of interactions among coexisting and interacting populations. If, as in the present study, multiseasonal and -annual data is lumped into one 'community' it is very questionable to what extent any metric derived from this data relates to processes that shape communities. Rather, an average signal from different communities living under contrasting env conditions is created. Interestingly, the authors suggest that different successional stages result in different scaling relationships in their tree data, but ignore this problem on the phytoplankton.

Minor comments

- For the analysis of phytoplankton, different fixatives were used (Lugol, Formaldehyde). It is known that Lugol results in considerable shrinking of cell size. This needs to be addressed in the calculation of cell sizes

- In Fig. 1, lines with several different colors are shown, but only one is explained in the caption.

Reviewer #2 (Remarks to the Author):

This manuscript compares the scaling of abundance (frequency of individuals) with body mass (in size classes) between aquatic phytoplankton and terrestrial tree communities. The study is based on the hypothesis of energetic equivalence holding that the allometric scaling of abundance should be inverse to that of metabolism. The main finding is that phytoplankton communities exhibit a scaling that is consistent with this expectation, whereas trees show a shallower scaling. Overall, the manuscript is very well written and the hypothesis is clearly described. The amount of data acquired for testing this hypothesis is impressive and the statistical analyses are highly appropriate. I have enjoyed reading this manuscripts, and there are only a few points mentioned below that might be worth considering.

First, the differences between tree and phytoplankton communities are discussed concerning ecosystem characteristics. However, one other difference between the two data sets is that phytoplankton samples include the entire autotrophic community of pelagic ecosystems, whereas trees are only a subset of the terrestrial autotrophs that also include (at least) herbs, grasses and mosses. I understand that resampling these communities is not an option, but a sensitivity analysis (preferred) or discussion of the consequences would be important.

Second, the underlying hypothesis is based on the allometric scaling of metabolism. The authors assume a three-quarter power-law scaling of metabolism for plants, but other exponents or non-linear scaling relationships have also been published (at least for animals). Hence, I think it is critically important for the conclusions of this study to present allometric scaling relationships for phytoplankton species (I think that Gabriel Yvon-Durocher might have published this) and trees (which might be found in Brian Enquist's work). Otherwise, it cannot be ruled out that the metabolism of trees follows an allometric power law with an 0.5 exponent and both groups are

thus consistent with energetic equivalence.

Third, in the description of the phytoplankton samples, I could not find whether and how mixotrophs, other unicells and detritus were excluded (I am sorry if I have overseen something). This should be described in detail as they could have a substantial effect on the abundance-mass scaling. In particular, mixotrophs or heterotrophic unicells could have a very different scaling relationship as they have different energy sources and thus violate the assumptions of the energetic equivalence hypothesis.

Fourth, I have to admit that I am struggling with the title. It claims that a general scaling law underpins the size structure of unicellular and multi-cellular autotroph communities. However, the results suggest that there are substantial differences between the scaling of these two community types. Please explain which generality you see across these two communities.

Fifth, is there any effect of the size of the body-mass bins on the scaling exponent? It would be great to see a short discussion and a sensitivity analysis on this.

Reviewer #3 (Remarks to the Author):

Review of Perkins et al., A general scaling law underpins the size-structure of unicellular and multicellular autotroph communities. Submitted to Nature Communications.

The authors present an analysis of size spectra/individual size distributions (ISDs) in trees and phytoplankton. They found that ISDs were consistent among tree communities and among phytoplankton communities but differed between trees and phytoplankton. ISDs of both trees and phytoplankton were shown to follow power laws but with different exponents. The authors suggest that differences in metabolic scaling between trees and phytoplankton might explain differences in power law exponents, with phytoplankton matching volumetric scaling ($-3/4$ power scaling of ISDs) and trees following diameter scaling ($-1/2$ power scaling of ISDs). They conclude that a universal scaling law might explain size scaling of primary producers.

This is a useful and interesting comparison of two taxonomic groups and draws on several large data sets. This is an unashamedly empirical study that is likely to stimulate further development of metabolic scaling theory at population and community levels. I have no major concerns with the methods or conclusions. The authors state that they will provide a pre-processed data set and code on publication.

Comments:

1. It might be a bit of a stretch to call this a comparison of unicellular and multicellular autotroph communities or a comparison of terrestrial and aquatic realms (e.g. title, l. 52, l. 88, l. 197). It's two taxonomic groups (trees and phytoplankton). The authors are generally careful to refer to trees and phytoplankton but I would suggest at least updating the title to reflect this point.

2. It's not entirely clear how much noise there is in the fitted exponents. I would like to see some quantification of absolute model fit (not just AIC), and perhaps a brief comment about potential extensions to the analytical method (e.g., a mixed/hierarchical model or a functional data analysis approach). As often stated in the macroecological literature, the noise around the general predictions is potentially more interesting than the general predictions themselves.

3. What is the role of non-metabolic mechanisms in determining size scaling in these systems? The authors touch on this in ll. 172-178 but I wonder whether processes such as disturbance, density dependence, and environmental heterogeneity are more important than currently stated? This discussion could build on a clearer quantification of the noise in the estimated exponents (see point 2) and might provide additional insight into the identification, analysis, and comparison of power laws among diverse realms (i.e. are universal consistencies in ISDs being masked by environmentally derived noise?).

We provide a point-by-point response to reviewers' comments below.

Reviewers' comments:

Reviewer #1 (Remarks to the Author):

Perkins et al compare allometric scaling relationships between phytoplankton and tree data. The paper is generally well written, but there seems to be a severe conceptual problem in terms how the two very different biomes are compared with regard to the life span of their organisms.

The scaling relationships addressed here are emergent properties of processes at the level of ecological communities (e.g. resource competition among competing taxa is one such key process). “a community is a group or association of populations of two or more different species occupying the same geographical area and in a particular time” (from wikipedia for convenience). Phytoplankton communities change rapidly due to short generation times (app 1 day) and fast dynamics of their environment (consider e.g. difference between spring bloom and summer stratification situation, see PEG model, Sommer et al). The community is an emergent property of interactions among coexisting and interacting populations. If, as in the present study, multiseasonal and -annual data is lumped into one ‘community’ it is very questionable to what extent any metric derived from this data relates to processes that shape communities. Rather, an average signal from different communities living under contrasting env conditions is created. Interestingly, the authors suggest that different successional stages result in different scaling relationships in their tree data, but ignore this problem on the phytoplankton.

We think that the reviewer has misunderstood our analyses and we are sorry for not expressing this more clearly. We agree entirely with the theory outlined by the reviewer and it is for this reason that we carried out two types of analysis with the phytoplankton data. The first, which the referee suggests as the most appropriate, treats each sample of the community at a given time point as a snap shot of the individuals that are coexisting at a particular point in time. We fit the different individual size distribution models to these data and compare to the tree analyses. However, as also noted by the reviewer, we say that (see page 9, line 5): “*The turnover of phytoplankton community composition is rapid in response to environmental variability owing to their small size and high capacity for dispersal.*” Therefore, we argue: “*point measurements of the ISD at a given location only provide*

a snapshot of community structure, which may deviate from steady-state depending on the local disturbance history.” Consequently, we carry out an additional analysis that pools all point measurements at a given location to give a time-integrated sample of the community. Interestingly, the mean among-site exponent derived across the point measurements for each location ($\bar{\lambda} + 1 = -0.85$; 95% confidence interval: -1.06 to -0.63) does not differ to the mean exponent after temporal-aggregation ($t = -1.26$, $df = 10$, $P = 0.237$). Our general finding that the mean ISD exponent ($\bar{\lambda} + 1$) for phytoplankton approximates -0.75 is therefore robust to whether or not temporal-aggregation is performed.

We appreciate that the reviewer may have missed the details of these analyses because the point measurements were in the supplementary information in the previous version of the manuscript. We have now moved the key results from the analysis of point measurements in the supplementary into the main text for clarity. Page 9, line 15: *“The ISD exponents after temporal aggregation were statistically indistinguishable from the ISD exponents derived from point measurements ($t = -1.26$, $df = 10$, $P = 0.237$; Supplementary Fig. 2), or from exponents observed for the spatial phytoplankton surveys ($t = -1.18$, $df = 93$, $P = 0.239$).”*

We have amended the methods text for clarity. Page 9, line 9:

“In order to test for the effects of temporal variation, we leveraged extensive time-series data of four marine and two limnetic freshwater stations. We aggregated data over multiple years (Supplementary Table 1) to build up a picture of the temporal ‘average’ composition of phytoplankton communities (Methods). We compared these temporally aggregated ISD exponents to the average exponents based upon all point measurements of the ISD within each station (Fig. S2).”

We have also modified the legend to Supplementary Fig. 2: *“Frequency histogram of individual size distribution exponents from point measurements of phytoplankton size structure. The mean among-site ISD exponent for the six stations ($\bar{\lambda} + 1 = -0.85$; 95% confidence interval: -1.06 to -0.63) does not differ from the mean among-site ISD exponent after temporal-aggregation (see Results in main text) or ISD exponents from spatially distributed communities ($t = -0.49$, $df = 93$, $P = 0.627$). Thus the stations included in the temporal data set can be considered representative of stations in the spatial data set and the general form of the ISD is invariant with temporal scale.”*

Minor comments

- For the analysis of phytoplankton, different fixatives were used (Lugol, Formaldehyde). It is known that Lugol results in considerable shrinking of cell size. This needs to be addressed in the calculation of cell sizes

We agree that changes in cell size (shrinkage and swelling) as a consequence of fixing samples with Lugol’s solution can occur (e.g. Stoecker *et al.*, 1994; Montagnes *et al.*, 1994; Mender-Deuer *et al.*, 2001). However, correction factors are highly variable between groups and can range from ca. 30% shrinkage to ca. 30% swelling for diatoms and dinoflagellates (Mender-Deuer *et al.*, 2001). Group-specific shrinkage correction factors are difficult to extrapolate to diverse natural phytoplankton communities because this information is not uniformly available for all groups thus we refrain from applying a correction for shrinkage in the analysis. Furthermore, correcting for the effects of preservative can be negligible: biomass estimates for

samples containing several species do not differ significantly whether based on live or fixed cell volumes (Mender-Deuer *et al.*, 2001). We therefore assume any shrinkage effect will be negligible in our analysis given the 4-6 orders of magnitude variation in cell volume we observe in the phytoplankton communities at a given location (Fig. 1). Finally, the preservation of samples from each site followed a standardized protocol so the exponent of the ISD for any given site will not be affected by fixative method because all individuals are preserved in the same way. We have added additional text in the Methods to highlight these points.

Page 15, line 12: *“The use of Lugol’s solution can result in changes in the size (shrinkage and swelling) of phytoplankton cells⁵⁷⁻⁵⁹. However, such effects are highly variable between groups⁵⁸ and extrapolating correction factors to diverse natural phytoplankton communities is problematic given the lack of general factors that are valid for naturally occurring plankton communities⁵⁹. Given the preservation of samples followed a standardised protocol for all datasets, we do not correct for the effects of preservation: we assume any effect will be negligible⁵⁸ given the 4-6 orders of magnitude variation in phytoplankton cell volume in this study (Fig. 1).”*

- In Fig. 1, lines with several different colors are shown, but only one is explained in the caption.

We have amended the text in the legend to clarify this. Page 5, line 8: *“The bounded power-law (magenta fitted line) was generally the best-supported ISD for tree and phytoplankton communities (Table 1), out-performing the (unbounded) power law or power-exponential models (turquoise and blue fitted lines, respectively).”*

Reviewer #2 (Remarks to the Author):

First, the differences between tree and phytoplankton communities are discussed concerning ecosystem characteristics. However, one other difference between the two data sets is that phytoplankton samples include the entire autotrophic community of pelagic ecosystems, whereas trees are only a subset of the terrestrial autotrophs that also include (at least) herbs, grasses and mosses. I understand that resampling these communities is not an option, but a sensitivity analysis (preferred) or discussion of the consequences would be important.

We thank the reviewer for raising this important point. The dataset used in the analysis of the phytoplankton size distributions only include nano- and micro-phytoplankton, and thus picoplankton (including cyanobacteria) are excluded because they cannot be reliably counted on a standard microscope set up. Hence we do not include the entire autotrophic community of pelagic systems in our analyses in much the same way that grasses and herbs are absent from the terrestrial data. Nonetheless, this is an important issue with an unknown influence on the size distribution. The referee is correct that these are limitations of the data currently available in the literature and it would require an entirely new research programme spanning many

decades to resample all of these sites to include missing groups. We have added text acknowledging these limitations to the discussion:

New text, page 13, line 17:

“Finally, it is important to consider that other taxonomic groups, not studied here, contribute to the total autotrophic production in forest and pelagic ecosystems. In forests, shrubs and herbaceous plants are abundant at the lower end of the size spectrum³⁹ whilst picophytoplankton (including cyanobacteria and picoeukaryotes) contribute significantly to primary production in pelagic ecosystems, especially in oligotrophic waters⁴⁰. We are unaware of comparable large-scale datasets comprising the size and abundance of shrubs and herbaceous plants so it is uncertain if the general scaling pattern we observe for trees > 1 cm extends down the size spectrum to these groups. Some evidence suggests that the individual size distribution for the entire autotrophic component of forests could be a discontinuous function with few species filling the size gap between shrubs and trees³⁹. In oligotrophic pelagic systems where cyanobacteria are abundant, a transition towards a steeper decline in abundance with body size might be expected²⁴ given the super-linear size-metabolism scaling for prokaryote species^{41,42} and equivalence in energy use among body mass classes.”

Second, the underlying hypothesis is based on the allometric scaling of metabolism. The authors assume a three-quarter power-law scaling of metabolism for plants, but other exponents or non-linear scaling relationships have also been published (at least for animals). Hence, I think it is critically important for the conclusions of this study to present allometric scaling relationships for phytoplankton species (I think that Gabriel Yvon-Durocher might have published this) and trees (which might be found in Brian Enquist’s work). Otherwise, it cannot be ruled out that the metabolism of trees follows an allometric power law with an 0.5 exponent and both groups are thus consistent with energetic equivalence.

We now present the exponents of allometric scaling of metabolism in the Introduction and refer to them again the Discussion.

New text, page 3, line 13:

“Since metabolic rates tend to scale as $M^{3/4}$ for large vascular plants^{15,17-19} and eukaryotic algae^{15,19-22}, the theoretical expectation is that the ISD follows a power-law with an exponent approximating $-3/4$ in both tree and phytoplankton communities.”

New text, page 11, line 6:

“Metabolism tends to scale with body mass with an exponent approximating $3/4$ for vascular plants (ranging from 0.1 to 100 cm in diameter^{18,31}). Although a larger exponent of ~ 1 is found among seedlings and saplings³², we are unaware of any data that supports a body mass-metabolism scaling exponent for trees of ~ 0.5 .”

New text, page 10, line 16:

“The theoretical three-quarter power scaling of metabolism with body mass for phytoplankton is in agreement with a number of empirical studies^{15,19-22} but contrasts with some studies that have reported isometric scaling exponents for phytoplankton metabolic rates both in the field^{24,29} and in the laboratory³⁰. These discrepancies

likely arise, in part, because the latter studies included smaller size fractions (picoplankton, which include prokaryotic autotrophs), than that studied here: when only species larger than $100 \mu\text{m}^3$ are considered (approximately the lower bound of the fitted distributions in this study; Fig. 1), the scaling of metabolism approximates $3/4^{30}$.

Third, in the description of the phytoplankton samples, I could not find whether and how mixotrophs, other unicells and detritus were excluded (I am sorry if I have overseen something). This should be described in detail as they could have a substantial effect on the abundance-mass scaling. In particular, mixotrophs or heterotrophic unicells could have a very different scaling relationship as they have different energy sources and thus violate the assumptions of the energetic equivalence hypothesis.

We than thank the reviewer for raising this important point. Heterotrophs and detritus are not included in any of the datasets as the primary aim of the surveys was to characterise the main autotrophic component of plankton communities (nano- and microphytoplankton). We have revised the text in Methods to clarify this. Page 15, line 20: “*Phytoplankton community data (nano- and microphytoplankton, excluding heterotrophs and detritus) included 92 open ocean stations, 75 from the Atlantic Meridional Transects (AMT 1–3; ⁵³) and 17 from additional Atlantic Ocean stations*”

With regards to mixotrophy, we appreciate that this could have some influence on the size structure of phytoplankton communities. Indeed recent theoretical work by Ward and Follows (2016) *PNAS*, suggests that mixotrophy could result in shallower size spectra and larger mean body size at the whole food web level owing to increases in the efficiency of trophic energy transfer through food chains.

Demonstrating mixotrophy by phytoplankton in natural planktonic communities is however a significant challenge and evidence about the physiology and mixed nutrition of these species comes predominately from work with cultures. Whilst mixotrophy has been assessed using the uptake of fluorescently labeled or radiolabeled prey (usually bacteria) to demonstrate phagotrophic activity in chloroplast-bearing protists (e.g. Zubkov and Tarran 2008, *Nature*), these approaches were beyond the scope of the plankton surveys performed in this study.

For two of the aquatic datasets (Lake Constance and Müggelsee) where high-resolution taxonomic identification was performed and where the presence of chloroplasts could be confirmed during routine microscopy, we have assessed the presence of ‘evidently mixotrophic’ taxa within the datasets (for these or closely related species, measurements of bacterivory are available in the literature). The contribution of evidently mixotrophic taxa make up only a small fraction, 14-16%, of the total number of individuals sampled within these datasets. However, mixotrophs can be conceptualized as existing along a continuum from nearly pure phototrophy to nearly pure heterotrophy, but a given species or morphotype’s position on this continuum can vary as result of various environmental factors – particularly nutrient availability. Consequently, any systematic classification of taxa in our datasets according to the literature would be plagued by uncertainty as a given taxon might potentially take different trophic strategies under different conditions. Thus, we are

unable to account for any putative impacts of mixotrophy in our analyses given the data we have at our disposal. That said, for the vast majority of chloroplast-bearing protists that contribute significantly to primary production in the oligotrophic ocean, energy metabolism is primarily driven by light and photosynthesis. Phagotrophy by these groups serves as an additional source of organic nutrients under conditions of extreme oligotrophy when inorganic nutrients are scarce. Consequently, all the taxa included in our analyses are primary producer – a small but variable fraction may access additional nutrition to fuel photosynthesis via phagotrophy – but ultimately they are primary producers that use sunlight to fix carbon at the base of the food chain.

Interestingly, the importance of mixotrophy for primary production is hypothesized to increase in the nutrient depleted oligotrophic regions of the ocean. When analyzing latitudinal variation in the ISD exponent in our marine dataset we find that more negative exponents (steep body mass–abundance scaling) occur in the oligotrophic regions owing to a predominance of small size classes and few large individuals. This finding contrasts with expectations that mixotrophy should lead to shallow size spectra and larger individuals. We hypothesise that the convergence in the size scaling exponent of phytoplankton communities with expectations based on energetic equivalence suggest that these primary producer communities are primary fueled by sunlight and that whilst mixotrophy may allow some members to access additional nutrient pools under certain conditions it does not systematically violate the assumptions of energy equivalence based on common energy source (e.g. sunlight).

We have added a paragraph to the Discussion highlighting these issues. New text, page 12, line 6:

“A key difference between tree and phytoplankton communities concerns mixotrophy, the ability of some photosynthetic protists to also ingest and assimilate living prey as a means of supplementing nutrient acquisition. Whilst largely absent in forests, mixotrophy is ubiquitous in pelagic ecosystems³⁴ and is suggested to exert a strong influence on the size structure of plankton communities at the global scale³⁵. We are unable to quantify any putative impacts of mixotrophy directly³⁶ in this study given the data we have at our disposal. However, for the vast majority of chloroplast-bearing protists that contribute significantly to primary production in pelagic ecosystems, energy metabolism is primarily driven by light and photosynthesis^{37,38}. Consequently, all of the taxa included in our analyses are primary producers – a small but variable fraction may access additional nutrition to fuel photosynthesis via phagotrophy when inorganic nutrients are scarce – but ultimately all use sunlight to fix carbon at the base of the food chain. Recent theoretical work³⁵ suggests that the mixotrophic contribution to primary production is greater in the nutrient depleted oligotrophic regions of the ocean and therefore we expected to observe less negative ISD exponents (i.e. shallower size spectra scaling) under oligotrophic conditions. In contrast, when analyzing latitudinal variation in the ISD exponent (Fig 2e), we find that more negative exponents occur in the oligotrophic oceanic regions (subtropical gyres) owing to a predominance of small size classes and few large individuals¹⁰. We hypothesise that the convergence in the size scaling exponent of phytoplankton communities with expectations based on energetic equivalence suggest that these primary producer communities are primary fueled by sunlight and that whilst mixotrophy may allow some members to access additional nutrient pools under certain

conditions it does not systematically violate the assumptions of energy equivalence based on common energy source (e.g. sunlight)."

Fourth, I have to admit that I am struggling with the title. It claims that a general scaling law underpins the size structure of unicellular and multi-cellular autotroph communities. However, the results suggest that there are substantial differences between the scaling of these two community types. Please explain which generality you see across these two communities.

We agree with the reviewer and have changed "a general scaling law" to "energetic equivalence" in the title to better reflect the main findings. New title: "*Energetic equivalence underpins the divergent size structure of tree and phytoplankton communities*"

Fifth, is there any effect of the size of the body-mass bins on the scaling exponent? It would be great to see a short discussion and a sensitivity analysis on this.

We use maximum likelihood estimation (MLE), rather than the logarithmic binning approach, which is one of the preferred approaches for estimating power-law exponents (White et al. 2008 *Ecology*, Edwards et al. 2016 *Methods in Ecology and Evolution*). MLE is a univariate method and finds the value of the exponent that maximizes the product of the probabilities of each observed body size value, x i.e., the product of $f(x)$ evaluated at each data point. This approach circumvents any issues with the size of body-mass bins (White et al. 2008 *Ecology*, Edwards et al. 2016 *Methods in Ecology and Evolution*).

Reviewer #3 (Remarks to the Author):

1. It might be a bit of a stretch to call this a comparison of unicellular and multicellular autotroph communities or a comparison of terrestrial and aquatic realms (e.g. title, l. 52, l. 88, l. 197). It's two taxonomic groups (trees and phytoplankton). The authors are generally careful to refer to trees and phytoplankton but I would suggest at least updating the title to reflect this point.

We have modified the title as suggested and refer to trees and phytoplankton, rather than terrestrial vs. aquatic realms or unicellular vs multicellular autotrophs, at various locations in the text: e.g. page 1, line 14; page 4, line 2; page 9, line 21, page 14, line 15; Table 1 & Fig. 2.)

2. It's not entirely clear how much noise there is in the fitted exponents. I would like to see some quantification of absolute model fit (not just AIC), and perhaps a brief comment about potential extensions to the analytical method (e.g., a mixed/hierarchical model or a functional data analysis approach). As often stated in the macroecological literature, the noise around the general predictions is potentially more interesting than the general predictions themselves.

3. What is the role of non-metabolic mechanisms in determining size scaling in these systems? The authors touch on this in ll. 172-178 but I wonder whether processes

such as disturbance, density dependence, and environmental heterogeneity are more important than currently stated? This discussion could build on a clearer quantification of the noise in the estimated exponents (see point 2) and might provide additional insight into the identification, analysis, and comparison of power laws among diverse realms (i.e. are universal consistencies in ISDs being masked by environmentally derived noise?).

Thank you for raising these points (outlined in comments 2 and 3).

We agree that there is a component of noise in our data, both in terms of how well the power law is supported and in terms of variability of the actual exponents. For each community we have now calculated the probability that a given empirical ISD differed from the theoretical distribution for each of the three power-law distributions from the KS statistic. We find that the empirical ISD was statistically indistinguishable from the theoretical distribution of the best-fitting power-law model in 87 % of tree communities and 83% of phytoplankton communities. While this indicates that there are a few communities for which the power-law family distributions do not entirely describe the data, we couldn't see any obvious pattern of ecological or geographical variation in the communities whose distribution was not well explained by the model. Some very large communities, comprising thousands of individuals (e.g. Barro Colorado Island; Fig 1), resulted statistically different from a power-law, but the origin of the mismatch was more statistical than real (with a large number of observations even small deviations from theoretical predictions can be detected) and the effect size was small. We have added these new results to main text.

New text, page 18, line 18:

“For each community, the probability that the empirical ISD differed from the theoretical distribution for the best-fitting power-law model was calculated from the KS statistic. It is important to note, however, with a large number of observations even small deviations from the theoretical distribution can be detected⁶⁴.”

New text, page 7, line 4:

“The empirical ISD was statistically indistinguishable from the theoretical distribution of the best-fitting power-law model in 87 % of tree communities and 83% of phytoplankton communities (Supplementary Data 1).”

New text, page 7, line 16:

“Estimates of the mean exponent were robust to the exclusion of sites where empirical ISDs differed from the theoretical distribution: trees -0.47 (95% confidence interval: -0.50 to -0.44) and phytoplankton -0.76 (95% confidence interval: -0.81 to -0.70).”

As we mention in the methods (page 19, line 6) we do not calculate the 95% confidence interval for the ISD exponent fitted to each community “since sample sizes varied between communities and hence the confidence intervals around fitted exponents would likely be wide when samples sizes were small, and consequently we would be more likely over-report the number of communities that follow the predictions (Duncanson et al. 2015).”

Independently of the point above, we agree with the reviewer that there is likely an important component of local variation across different communities, which affects for instance the exponent of the power law (see Figure 2 in the main text). We have added additional text to the discussion to highlight this.

New text, page 13, line 7:

“The primary aim of this study was to examine macroecological patterns in autotroph size structure; hence this study does not assess the role of local factors (such as disturbance, density dependence, and environmental heterogeneity etc.) on individual size distributions. The number of communities where empirical ISD relationships differed from theoretical power-law distributions was similar among groups (approx. 15%), as was the spread of ISD exponents around the mean (Fig. 2). These results suggest that non-metabolic processes might play an important role at a local scale^{9,11,23}, but these effects appear to be comparable across forest and pelagic ecosystems. Future studies should consider the residual variation around the macroecological patterns as it is likely to offer clues to the other factors, in addition to body size, that affect ecological processes in terrestrial and aquatic realms.”

REVIEWERS' COMMENTS:

Reviewer #2 (Remarks to the Author):

The revision has carefully addressed all of the points raised by my review. I think this is an excellent study and I have no further comments.

Reviewer #3 (Remarks to the Author):

I am satisfied that the authors have addressed the comments from my previous review (I was Reviewer 3).